# Amino Acid Profile Alterations in the Mother–Fetus System in Gestational Diabetes Mellitus and Macrosomia

**DOI:** 10.3390/ijms26178351

**Published:** 2025-08-28

**Authors:** Natalia. A. Frankevich, Alisa. O. Tokareva, Sergey. Yu. Yuriev, Vitaly. V. Chagovets, Anastasia. A. Kutsenko, Anastasia. V. Novoselova, Tamara. E. Karapetian, Vadim. V. Lagutin, Vladimir. E. Frankevich, Gennady. T. Sukhikh

**Affiliations:** 1National Medical Research Center for Obstetrics Gynecology and Perinatology Named After Academician V.I. Kulakov, Ministry of Healthcare of Russian Federation, 117997 Moscow, Russia; a_tokareva@oparina4.ru (A.O.T.); v_chagovets@oparina4.ru (V.V.C.); a_novoselova@oparina4.ru (A.V.N.); t_karapetyan@oparina4.ru (T.E.K.); v_lagutin@oparina4.ru (V.V.L.); v_frankevich@oparina4.ru (V.E.F.); g_sukhikh@oparina4.ru (G.T.S.); 2Department of Obstetrics and Gynecology, Siberian State Medical University, 634050 Tomsk, Russia; yurev.sy@ssmu.ru (S.Y.Y.); kutsenko.aa@ssmu.ru (A.A.K.); 3Laboratory of Translational Medicine, Siberian State Medical University, 634050 Tomsk, Russia

**Keywords:** amino acids, metabolic profile, gestational diabetes mellitus, clinical markers, biological markers, macrosomia, mother–fetus system

## Abstract

Gestational diabetes mellitus (GDM) is a growing global health concern, driving the need for novel diagnostic and prognostic approaches. The aim of this study was to analyze the amino acid profile in the mother–fetus system (maternal venous blood, umbilical cord blood, and amniotic fluid) and to identify specific biological markers of GDM and macrosomia. Using HPLC-MS/MS, we analyzed serum from maternal venous and umbilical cord blood, along with amniotic fluid, across 94 mother–fetus pairs (53 GDM, 41 controls). Machine learning and metabolic pathway analysis revealed significant alterations in 19 amino acids. In GDM, maternal serum showed elevated 5-OH-lysine and homocitrulline, while cord blood had higher isoleucine, serine, and threonine. Amniotic fluid exhibited increased leucine, isoleucine, threonine, serine, arginine, and ornithine. Conversely, histidine, glutamine, alanine, asparagine, β-/γ-aminobutyric acids, phenylalanine, ornithine, and citrulline were reduced. Histidine, glutamine, and asparagine inversely correlated with blood glucose (r = −0.26, r = −0.33, r = −0.30) and were lower in GDM. These findings highlight three key metabolic loci in GDM pathogenesis, with glutamine, histidine, and asparagine emerging as potential maternal blood biomarkers for early macrosomia prediction. However, given confounding factors in metabolomic studies, further large-scale validation is essential.

## 1. Introduction

Gestational diabetes mellitus (GDM) and associated fetal macrosomia are becoming increasingly significant global health challenges, necessitating the development of new diagnostic and prognostic approaches. The global standardized prevalence of GDM is 14.0% (95% CI: 13.97–14.04%), with regional variations ranging from 7.1% to 27.6% in 2022 [1]. Among pregnant women with GDM, fetal macrosomia is diagnosed in 15–45% of cases—three times more frequently than in mothers with normal blood glucose levels. According to WHO criteria, fetal macrosomia is defined as a condition where the estimated or actual fetal/neonatal body weight exceeds 4000 g (regardless of gestational age) (WHO guidelines on GDM, 2013). Perinatal mortality rates are 1.5–3 times higher in macrosomic infants compared to those with normal birth weight [2,3,4], and the risk of intrapartum complications increases significantly [5]. Mothers with GDM and fetal macrosomia are more likely to experience complications during delivery and the postpartum period, with a substantially higher rate of cesarean sections [2,3,4,6]. Neonatal morbidity rates are also elevated in macrosomic newborns (birth weight > 4000 g) [5]. This cohort of children is at high risk for developing type 2 diabetes, obesity, and cardiovascular diseases in adolescence and adulthood [7].

Amino acid profiling has demonstrated metabolic shifts in pregnant women with elevated insulin resistance and hyperglycemia, mirroring patterns observed in non-pregnant individuals with insulin resistance [8]. This suggests shared metabolic pathways underlying insulin resistance and hyperglycemia, irrespective of pregnancy status.

Most published studies focus solely on maternal metabolic changes. A comprehensive analysis of the maternal–fetal system—integrating maternal venous blood, amniotic fluid, and umbilical cord blood—could unify existing multiomics data and advance a systems biology approach to address GDM.

In GDM, the fetomaternal unit exhibits significant disruptions in carbohydrate and lipid metabolism [5]. Beyond glucose and lipid metabolic alterations, GDM and associated macrosomia are characterized by an imbalance in “free” amino acid levels. In healthy pregnancies, a series of enzymatic reactions maintain dynamic equilibrium in the fetomaternal system—replenishing intermediate metabolites through anaplerotic reactions while balancing their utilization and removal from metabolic cycles.

GDM alters both amino acid composition and associated anaplerotic pathways in the mother–fetus system. The amino acid profile undergoes marked changes: decreased levels of essential amino acids, elevated concentrations of aromatic (phenylalanine, tyrosine) and sulfur-containing amino acids (methionine, cysteine), and altered ratios of branched-chain amino acids (leucine, isoleucine, valine). These shifts reflect placental dysfunction, insulin resistance, modified activity of amino-acid-metabolizing enzymes, and increased fetal demand.

Consequently, investigating amino acid amphibolism (intermediary metabolic reactions) in GDM is critical—not only to elucidate disease pathogenesis but also to enable early diagnosis of complications and develop personalized metabolic interventions tailored to maternal dietary habits and lifestyle.

The aim of this study was to analyze the amino acid profile in the mother–fetus system (maternal venous blood, umbilical cord blood, and amniotic fluid) to identify specific biological markers of GDM and macrosomia.

## 2. Results

At the initial stage of the study, a prospective selection was made from a cohort of 2000 women who underwent first-trimester prenatal screening (11–13.6 weeks) at the Tomsk Perinatal Center. This selection resulted in 94 mother–newborn pairs, who were divided into two main groups: Group I (patients with GDM, *n* = 53) and control Group II (non-GDM control group, *n* = 41).

Subsequent stratification identified three clinically significant subgroups: Ia (GDM with macrosomia, ≥4000 g, *n* = 23), Ib (GDM with normal birth weight, 2501–3999 g, *n* = 30), and II (non-GDM control group, 2501–3999 g, *n* = 36 after the exclusion of 5 cases exceeding 3999 g).

The study received ethical approval, and all participants provided written informed consent prior to their inclusion in the research. The groups were comparable in age, gestational age at delivery, and fetal sex (Appendix A). Mean maternal age was 33.5 years in the GDM group versus 34 years in controls. All participants delivered at 39 weeks of gestation (38; 39). Neonatal sex distribution showed no significant intergroup difference (*p* = 0.49). Prepregnancy BMI was lower in controls (22 kg/m^2^). Recent evidence identifies prepregnancy BMI as a key clinical predictor of GDM development. Notably, women with BMI 22 kg/m^2^ (21; 24) who had comparable total weight gain to the GDM-with-macrosomia subgroup (13 kg (11; 16) vs. 14 kg (11; 16), respectively) did not develop GDM.

Of particular interest, the lowest weight gain occurred in the GDM subgroup without macrosomia (11 kg (9; 14), *p* < 0.02), highlighting the significant impact of dietary control and lifestyle modifications on preventing macrosomia in GDM.

Cesarean delivery rates were significantly higher in GDM women versus controls (*p* = 0.003), reaching 70% in the macrosomia subgroup. Ultrasound evaluation revealed significant increases in placental thickness and amniotic fluid pocket depth in GDM patients with macrosomia (*p* = 0.007 and *p* = 0.03, respectively).

### 2.1. Venous Blood Serum

Based on mutual information index values, the following potential biomarkers of GDM were identified: lysine, glutamine, histidine, alanine, γ-aminobutyric acid, β-alanine, valine, homocitrulline, asparagine, 5-OH-lysine, tryptophan, and α-aminobutyric acid (Figure 1a). The levels of 5-OH-lysine and homocitrulline were elevated in GDM (FC = 7.2, *p* = 0.05 and FC = 4.3, *p* = 0.94, respectively), whereas the concentrations of α-aminobutyric acid (FC = 0.003, *p* = 0.32), β-alanine (FC = 0.004, *p* = 0.14), γ-aminobutyric acid (FC = 0.54, *p* = 0.04), glutamine (FC = 0.59, *p* = 0.04), lysine (FC = 0.44, *p* = 0.03), alanine (FC = 0.37, *p* = 0.01), asparagine (FC = 0.49, *p* = 0.01), and histidine (FC = 0.20, *p* = 0.003) were reduced (Figure 1b).

In the control group, Shapley additive explanations (SHAP, measure of the effect to the result) value analysis revealed decreased levels of 5-OH-lysine and increased levels of glutamine, histidine, and asparagine (Figure 2). These same amino acids were identified as potential biomarkers for distinguishing between control and GDM groups. In GDM with macrosomia, we observed decreased levels of asparagine (FC = 0.39) and lysine (FC = 0.44), along with elevated levels of homocitrulline (FC = 4.9) and 5-OH-lysine (FC = 2.1). Also, the level of asparagine in the case of GDM with macrosomia was statistically significantly lower than in the control group (*p* = 0.01, Appendix A). Notably, while histidine effectively differentiates controls from GDM cases, it does not serve as a reliable biomarker for the GDM+macrosomia subgroup. For normosomic GDM cases, lower levels of histidine (FC = 0.27) and glutamine (FC = 0.82) were characteristic, accompanied by higher levels of 5-OH-lysine (FC = 3.5) and serine (non-zero levels only in normasomia cases). Histidine, asparagine, and glutamine have high sensitivity for GDM detection and high specificity for macrosomia detection (Table 1). These findings suggest that: (1) decreased asparagine levels are associated with GDM complicated by macrosomia; (2) reduced histidine and glutamine levels may be linked to GDM without macrosomia; and (3) elevated 5-OH-lysine levels are common to both GDM subgroups.

Analysis of metabolic pathway databases identified 45 statistically significant pathways enriched with amino acids differentiating control and GDM groups (maternal venous blood). The most relevant pathways included: (1) “Amino acid transport across plasma membrane” and “Transport of inorganic cations/anions and amino acids/oligopeptides” (β-alanine, alanine, γ-aminobutyric acid, glutamine-enriched); (2) “Na+-dependent neurotransmitter transporters” and “Neurotransmitter release cycle” (β-alanine, γ-aminobutyric acid, glutamine-enriched); (3) “GABA reuptake” and “GABA synthesis/release/reuptake/degradation” (β-alanine, γ-aminobutyric acid-enriched); (4) Reactome’s “Aspartate/asparagine metabolism” (β-alanine, glutamine-enriched); (5) KEGG’s “GABAergic synapse” and “Alanine/aspartate/glutamate metabolism” (glutamine, γ-aminobutyric acid-enriched); and (6) KEGG’s “Beta-alanine metabolism” (β-alanine, γ-aminobutyric acid-enriched). The pathways have FDR < 0.01 (Figure 3).

No statistically significantly enriched pathways were identified for the GDM+macrosomia subgroup biomarkers.

### 2.2. Serum Cord Blood

Analysis of umbilical cord serum identified phenylalanine, histidine, ornithine, citrulline, serine, α-aminobutyric acid, and isoleucine as potential GDM biomarkers (Figure 4a). Among these, isoleucine (FC = 1.1, *p* = 0.01) showed a statistically significant increase in GDM cases. Serine (FC = 1.01, *p* = 0.35) and α-aminobutyric acid (FC = 1.08, *p* = 0.12) levels were also elevated, whereas phenylalanine (FC = 0.97, *p* = 0.22), histidine (FC = 0.94, *p* = 0.09), ornithine (FC = 0.94, *p* = 0.26), and citrulline (FC = 0.92, *p* = 0.09) levels were reduced in umbilical cord blood when the mother had GDM (Figure 4b).

The control group exhibited lower levels of isoleucine (FC = 0.83) and serine (FC = 0.98), but higher levels of methionine (FC = 1.1), citrulline (FC = 1.1), proline (FC = 1.1), lysine (FC = 1.05), phenylalanine (FC = 1.03), and histidine (FC = 1.05). The GDM+macrosomia subgroup showed reduced lysine (FC = 0.91) levels and elevated threonine (FC = 1.2), isoleucine (FC = 1.2), and serine (FC = 1.1) levels, while the GDM+normosomia subgroup displayed lower methionine (FC = 0.90), citrulline (FC = 0.84), threonine (FC = 0.98), and proline (FC = 0.97) levels but higher tryptophan (FC = 1.1), isoleucine (FC = 1.07), and arginine (FC = 1.09) levels. Also, level of isoleucine in GDM+macrosomia subgroup is statistically significantly higher than in the control group (*p* = 0.02, Appendix A). Notably, histidine served as a biomarker for controls in both umbilical cord blood and maternal serum, whereas lysine was specific to GDM+macrosomia in both sample types. No common biomarkers were found between umbilical cord blood and maternal serum for GDM+normosomia. Importantly, threonine emerged as a discriminant marker between GDM subgroups based on fetal macrosomia status, with low lysine and high serine levels being specifically characteristic of GDM+macrosomia (Figure 5).

No statistically significant pathways enriched with umbilical cord blood amino acid biomarkers of GDM were identified.

A total of 46 pathways were found to be statistically significantly enriched with umbilical cord blood amino acids serving as biomarkers for “GDM in pregnancy + fetal macrosomia”. The most significant pathways included: (1) “tRNA charging, gamma-glutamyl cycle”, (2) “leukotriene biosynthesis”, and (3) “S-methyl-5-thio-alpha-D-ribose 1-phosphate degradation” from HumanCyc; (4) “Mineral absorption—Homo sapiens (human)” from KEGG (all enriched with isoleucine, serine, and threonine); as well as (5) pathways of tobramycin, streptomycin, paromomycin, gentamicin, and chloramphenicol action from SMPDB (enriched with isoleucine and threonine). The pathways have FDR < 0.001 (Figure 6).

### 2.3. Amniotic Fluid

Analysis of amniotic fluid amino acids identified potential biomarkers of GDM: ornithine, serine, leucine, citrulline, tyrosine, proline, phenylalanine, isoleucine, glycine, threonine, glutamine, asparagine, arginine, valine, 1-methylhistidine, aspartic acid, alanine, glutamic acid, and histidine (Figure 7a). Among these, the levels of ornithine (FC = 1.4, *p* = 0.01), serine (FC = 1.6, *p* = 0.02), leucine (FC = 1.6, *p* = 0.004), citrulline (FC = 1.4, *p* = 0.045), tyrosine (FC = 1.3, *p* = 0.04), proline (FC = 1.6, *p* = 0.02), phenylalanine (FC = 1.4, *p* = 0.03), isoleucine (FC = 1.6, *p* = 0.004), threonine (FC = 1.5, *p* = 0.003), glutamine (FC = 1.2, *p* = 0.01), arginine (FC = 1.2, *p* = 0.007), valine (FC = 1.2, *p* = 0.007), aspartic acid (FC = 1.7, *p* = 0.007), alanine (FC = 1.2, *p* = 0.02), glutamic acid (FC = 1.5, *p* = 0.006), and histidine (FC = 1.2, *p* = 0.02) were significantly elevated in GDM cases (Figure 7b).

The control group exhibited lower levels of leucine (FC = 0.67), threonine (FC = 0.69), 5-OH-lysine (FC = 0.80), arginine (FC = 0.89), isoleucine (FC = 0.67), serine (FC = 0.68), aspartic acid (FC = 0.67), 3-methylhistidine (FC = 0.64), citrulline (FC = 0.71), proline (FC = 0.69), ornithine (FC = 0.73), histidine (FC = 0.94), and phenylalanine (FC = 0.81). With the exception of 5-OH-lysine and 3-methylhistidine, these amino acids were previously identified as potential biomarkers distinguishing controls from GDM cases. While isoleucine, serine, citrulline, proline, histidine, and phenylalanine were established as control group markers in umbilical cord blood, only isoleucine and serine showed consistent directional changes in both amniotic fluid and cord blood in GDM. Histidine and 5-OH-lysine were previously identified as control markers in serum, with only 5-OH-lysine showing consistent reduction in both serum and amniotic fluid. The GDM+macrosomia subgroup demonstrated elevated levels of serine (FC = 1.2), citrulline (FC = 1.4), 5-OH-lysine (FC = 1.1), leucine (FC = 1.4), isoleucine (FC = 1.2), threonine (FC = 1.1), and aspartic acid (FC = 1.1), alongside reduced γ-aminobutyric acid (FC = 0.88), phenylalanine (FC = 0.93), and ornithine (FC = 0.99). The increases in serine and threonine paralleled umbilical cord blood findings, while 5-OH-lysine elevation matched serum patterns. For normosomic GDM, characteristic elevations included arginine (FC = 1.2), 3-methylhistidine (FC = 1.6), ornithine (FC = 1.5), threonine (FC = 1.1), γ-aminobutyric acid (FC = 1.4), leucine (FC = 1.6), 5-OH-lysine (FC = 1.2), and isoleucine (FC = 1.4). Also, levels of arginine (*p* = 0.006), aspartic acid (*p* = 0.008), γ-aminobutyric acid (*p* = 0.02), and isoleucine (*p* = 0.006) were statistically significantly higher in the GDM+normosomia subgroup than in the control group (Appendix A). While arginine, threonine, and isoleucine in cord blood were also potential normosomic GDM markers, threonine showed opposite directional changes (decreased in cord blood vs. increased in amniotic fluid). The rise in 5-OH-lysine was consistent with serum findings. Notably, increased arginine, 3-methylhistidine, and ornithine were unique to normosomic GDM, as these either decreased in controls or showed non-significant changes in GDM+macrosomia. Conversely, serine elevation was specific to GDM+macrosomia (Figure 8).

A total of 305 pathways were found to be statistically significantly enriched with amniotic fluid biomarkers differentiating GDM from controls. The most significant pathways included: (1) “tRNA charging, gamma-glutamyl cycle”, (2) “leukotriene biosynthesis”, and (3) “S-methyl-5-thio-alpha-D-ribose 1-phosphate degradation” from HumanCyc; (4) “Protein digestion and absorption—Homo sapiens (human)” and (5) “Aminoacyl-tRNA biosynthesis—Homo sapiens (human)” from KEGG (enriched with glutamic acid, proline, leucine, tyrosine, isoleucine, arginine, serine, glutamine, phenylalanine, threonine, aspartic acid, and valine); (6) “Amino acid transport across the plasma membrane” and (7) “Transport of inorganic cations/anions and amino acids/oligopeptides” from Reactome (enriched with glutamic acid, proline, leucine, tyrosine, isoleucine, arginine, alanine, ornithine, glutamine, phenylalanine, aspartic acid, and valine); (8) “Central carbon metabolism in cancer—Homo sapiens (human)” from KEGG (enriched with glutamic acid, proline, phenylalanine, tyrosine, isoleucine, arginine, serine, glutamine, leucine, aspartic acid, and valine); and (9) “Glucose Homeostasis” from WikiPathways (enriched with phenylalanine, tyrosine, arginine, isoleucine, citrulline, ornithine, and valine). The pathways have FDR < 0.01 (Figure 9).

Analysis identified 134 statistically significant pathways enriched with amniotic fluid biomarkers for the GDM+macrosomia subgroup. Key pathways included: (1) “tRNA charging” (2) “leukotriene biosynthesis”, and (3) “gamma-glutamyl cycle S-methyl-5-thio-alpha-D-,ribose 1-phosphate degradation” from HumanCyc; (4) “Protein digestion and absorption—Homo sapiens (human)” and (5) “Aminoacyl-tRNA biosynthesis—Homo sapiens (human)” from KEGG (enriched with phenylalanine, serine, isoleucine, leucine, threonine, and aspartic acid); (6) “Mineral absorption—Homo sapiens (human)” (enriched with leucine, serine, phenylalanine, isoleucine, and threonine) and (7) “Central carbon metabolism in cancer—Homo sapiens (human)” (enriched with serine, aspartic acid, phenylalanine, leucine, and isoleucine) from KEGG; (8) “Amino acid transport across the plasma membrane” from Reactome (enriched with leucine, isoleucine, γ-aminobutyric acid, phenylalanine, and ornithine); and (9) “Glucose Homeostasis” from WikiPathways (enriched with phenylalanine, isoleucine, ornithine, and citrulline). The pathways have FDR < 0.01 (Figure 10).

### 2.4. Clinical Markers

To validate the biological significance of the findings and assess potential biomarkers of GDM and macrosomia in maternal blood, we analyzed clinical-laboratory markers and established their correlation with experimental data (maternal blood amino acids).

Detailed clinical characteristics and routine laboratory/imaging data extracted from primary medical records are presented in Appendix A. The analysis revealed that maternal prepregnancy BMI and glucose levels showed the strongest association with GDM development (Figure 11a). The relationship between GDM and BMI followed a linear trend in the BMI range of 22.5–30 kg/m^2^, while for BMI > 35 kg/m^2^, the association plateaued (Figure 11b). Furthermore, parameters falling within Quadrants I and III of Figure 11c exhibited the most significant association with GDM: Quadrant I (BMI > 25 kg/m^2^ and glucose > 4.0 mmol/L) correlated with a high probability of GDM, whereas Quadrant III (BMI < 25 kg/m^2^ and glucose < 4.0 mmol/L) was associated with a low GDM risk.

The strongest associations with neonatal birth weight were observed for amniotic fluid pocket depth (AFP) and ultrasound markers of diabetic fetopathy (DF). Gestational age at delivery ranked seventh in predictive strength, following placental thickness and maternal levels of alanine aminotransferase (ALT), aspartate aminotransferase (AST), and fibrinogen. Notably, the type of GDM treatment (insulin therapy vs. dietary management) showed virtually no impact on fetal weight (Figure 12a). The relationship between AFI and birth weight followed a linear pattern, with AFI values ≥ 90 mm being consistently associated with DF diagnosis (Figure 12b).

Subsequent analysis revealed significant moderate-strength correlations between clinical parameters and maternal blood amino acids (|R| < 0.4, *p* < 0.05). Specifically, glucose levels showed weak inverse correlations with β-alanine (r = −0.28, *p* = 0.01), γ-aminobutyric acid (r = −0.31, *p* = 0.006), histidine (r = −0.26, *p* = 0.02), lysine (r = −0.28, *p* = 0.02), methionine (r = −0.25, *p* = 0.03), 1-methylhistidine (r = −0.25, *p* = 0.03), phenylalanine (r = −0.28, *p* = 0.02), asparagine (r = −0.29, *p* = 0.01), and glutamine (r = −0.33, *p* = 0.003). Similarly, BMI demonstrated weak negative correlations with proline (r = −0.33, *p* = 0.004), alanine (r = −0.26, *p* = 0.02), histidine (r = −0.28, *p* = 0.02), and γ-aminobutyric acid (r = −0.25, *p* = 0.03). APTT levels showed inverse associations with leucine (r = −0.24, *p* = 0.03), asparagine (r = −0.30, *p* = 0.008), glutamine (r = −0.23, *p* = 0.047), aspartic acid (r = −0.32, *p* = 0.008), and threonine (r = −0.27, *p* = 0.02). Conversely, creatinine exhibited weak positive correlations with γ-aminobutyric acid (r = 0.30, *p* = 0.008), histidine (r = 0.26, *p* = 0.03), lysine (r = 0.25, *p* = 0.03), and 1-methylhistidine (r = 0.29, *p* = 0.01) (Figure 13). Of particular interest was the observed relationship between clinical-laboratory parameters and amino acids previously identified as strongly associated with GDM and macrosomia in maternal venous blood, notably histidine (negative correlation: r = −0.26, *p* = 0.02).

The most pronounced alterations were observed in amino acids involved in glucose metabolism (e.g., glutamine, histidine) and markers of renal function (e.g., creatinine and associated amino acids). The correlation analysis results highlight the interplay between metabolic disturbances in gestational diabetes and serum amino acid profiles.

Thus, maternal blood amino acids—specifically asparagine, glutamine, and histidine, which showed strong associations with GDM (glutamine, histidine) and macrosomia (asparagine)—demonstrated inverse correlations with blood glucose levels, suggesting their biological significance as potential biomarkers in GDM pathophysiology.

## 3. Discussion

Our study revealed significant alterations in the amino acid profile within the mother–fetus system in GDM and associated fetal macrosomia. We identified key clinical markers consistent with recent international findings [9,10,11,12,13,14].

Analysis of maternal serum, umbilical cord blood, and amniotic fluid demonstrated specific metabolic shifts that may may contribute to the development of early diagnostic markers in the future, after validation in a large cohort of women in early pregnancy.

In maternal venous serum, GDM was characterized by decreased levels of histidine, glutamine, alanine, asparagine, and β- and γ-aminobutyric acids, reflecting potential impairments in amino acid transport and neurotransmitter metabolism. Elevated 5-OH-lysine and homocitrulline were GDM-associated, independent of macrosomia status. The most significantly affected metabolic pathways included amino acid transport, GABAergic synapse function, and alanine/aspartate metabolism. GABA-associated pathways are enriched by serum amino acids, whose levels were altered in the case of GDM. Therefore, GABA has an important role in glucose homeostasis. Neurotransmitter pathways have a strong association with diabetes development.

The study revealed significantly decreased concentrations of histidine, phenylalanine, ornithine, and citrulline in umbilical cord blood from GDM pregnancies, suggesting impaired fetal nitrogen metabolism. Elevated levels of isoleucine, serine, and threonine were characteristic of fetal macrosomia, consistent with enhanced fetal anabolic processes. These changes showed significant associations with protein breakdown/absorption, mineral metabolism, and leukotriene biosynthesis pathways, likely reflecting fetal adaptive mechanisms to maternal hyperglycemia. The antibiotics-related pathways, enriched by isoleucine and threonine (isoleucine’s precursor), can be associated with increasing error rate of including isoleucine in the amino acid sequences [15,16]. Furthermore, we identified key metabolic pathways involving amino acids in GDM pathophysiology.

In GDM pregnancies, amniotic fluid demonstrated significantly elevated levels of leucine, isoleucine, threonine, serine, arginine, ornithine, and 5-OH-lysine (*p* < 0.05), reflecting activated anabolic processes and altered protein metabolism. While serine and threonine emerged as potential macrosomia biomarkers (r = 0.32–0.41), arginine and ornithine were more characteristic of GDM without macrosomia. These changes were associated with three key metabolic pathways: (1) protein digestion/absorption (KEGG hsa04974), (2) aminoacyl-tRNA biosynthesis (KEGG hsa00970), and (3) amino acid transmembrane transport (GO:0003333), with all findings remaining significant after FDR correction (q < 0.01). tRNA-charging, leucotriene biosynthesis, gamma-glutamyl cycle are enriched by serum cord and amniotic fluid biomarkers of GDM and macrosomia. Proteins, which take part in tRNA-aminoacylation, is also involved in diabetes mellitus development [17]. A study using mice demonstrated that a higher level of leukotriene increased the level of insulin resistance [18]. Gamma-glutamyl transferase is an important element of the gamma-glutamyl cycle and its higher level is associated with higher levels of type 2 diabetes [19,20]. Also, higher activity of the tRNA-charging pathway and gamma-glutamyl cycle is associated with a higher level of protein synthesis and can be associated with macrosomia.

Independent predictors of GDM included prepregnancy BMI and glucose levels, while amniotic fluid index (AFI) and ultrasound signs of diabetic fetopathy after 37 gestational weeks were predictive of macrosomia. Women carrying macrosomic fetuses exhibited significantly greater gestational weight gain regardless of GDM status. Correlation analysis identified three maternal serum amino acids with biomarker potential for GDM and macrosomia: glutamine (inversely correlated with glucose levels, *p* < 0.01), histidine (associated with both GDM and renal function markers), and asparagine (macrosomia-specific pattern).

Table 2 provides a comparative overview of amino acid metabolism alterations in the mother–fetus system and identifies key biomarkers for further investigation (see Appendix A for complete quantitative data on all amino acids across study groups and subgroups).

Of particular interest was the analysis of publications examining the characteristics of the amino acid profile in GDM and a comparison of the obtained results. In a large-scale study by Zhen Hong et al. [21] involving 969 women, conducted from 2019 to 2021, the relationship between plasma amino acid concentrations and GDM incidence was investigated. The study identified 16 biomarkers differentiating GDM and non-GDM groups, revealing strong correlations (OR = 0.753–1.671, *p* = <0.001–0.001) between specific amino acids and GDM risk. Notably, GDM was associated with significantly elevated levels of alpha-aminoadipic acid and arginine, alongside reduced glycine and serine (*p* = <0.001–0.045). However, the authors concluded that the overall amino acid profile, rather than disruptions in individual amino acids, may represent a more significant preventive or therapeutic target for GDM [21]. In a 2022 study by Xiangju Kong et al. [22], clinical data and serum levels of fatty acids, amino acids, and organic acids were analyzed in 90 pregnant women at 24–28 weeks of gestation to compare differences among groups with gestational hypertension (GH), GDM, and healthy pregnancies. GDM was characterized by lower pyroglutamic acid and higher 2-hydroxybutyric and glutamic acid levels. The findings suggested that metabolic disorders (GDM and GH) in mid-pregnancy were associated with dysregulated glucose and lipid metabolism, potentially inducing oxidative stress through glutathione metabolism and unsaturated fatty acid biosynthesis [22]. Our study revealed elevated maternal venous blood concentrations of 5-OH-lysine and homocitrulline in GDM, while current evidence indicates an inverse relationship between lysine levels and blood triglycerides in metabolic disorders. Similarly, Sunmin Park et al. [23] demonstrated associations between lysine, tyrosine, and valine levels and GDM, insulin resistance, and insulin secretion at 24–28 weeks of gestation [23].

In 2017, Najmeh Rahimi et al. [24] published an analysis of venous blood amino acids (AAs) in women with gestational diabetes mellitus and type 2 diabetes (T2DM). Compared to controls, mothers with GDM exhibited higher plasma concentrations of arginine (*p* = 0.01), glycine (*p* = 0.01), and methionine (*p* = 0.04), whereas pregnant women with T2DM had elevated plasma levels of asparagine (*p* = 0.01), tyrosine (*p* < 0.01), valine (*p* < 0.01), phenylalanine (*p* < 0.01), glutamine (*p* < 0.01), and isoleucine (*p* < 0.01). Regression analysis confirmed significantly increased plasma concentrations of asparagine (OR: 3.64, CI 1.22–10.47), threonine (OR: 3.38, CI 1.39–8.25), aspartic acid (OR: 3.92, CI 1.19–12.91), phenylalanine (OR: 2.66, CI 1.01–6.94), glutamine (OR: 2.53, CI 1.02–6.26), and arginine (OR: 1.96, CI 1.02–3.76) in GDM mothers after adjusting for gestational age and BMI [24]. Several studies on amino acid profiles in GDM and T2DM have reported elevated levels of arginine, methionine, and glycine in women with GDM. In GDM, dysregulation of the adenosine/L-arginine/nitric oxide pathway likely occurs [25], which may underlie vascular endothelial dysfunction. The significant increase in arginine concentration observed in our study in amniotic fluid suggests potential fetal involvement in these processes and strongly indicates signs of endothelial dysfunction in GDM, even among fetuses without signs of diabetic fetopathy.

In a 2005 publication, the authors identified several biomarkers of GDM, including increased placental weight, a reduced fetal-to-placental weight ratio compared to normal pregnancies, significantly elevated ornithine levels in venous blood, and altered amino acid (AA) profiles in umbilical cord blood—specifically increased valine, methionine, phenylalanine, isoleucine, leucine, ornithine, glutamate, proline, and alanine, alongside decreased glutamine [26]. Of particular interest is the involvement of fetal AA pools (umbilical cord blood and amniotic fluid) in the gamma-glutamyl cycle, potentially linked to fetal macrosomia. In their 2005 study, Irene Cetin et al. [26] reported significantly elevated glutamate and reduced glutamine levels in umbilical cord blood of GDM pregnancies, suggesting increased fetal hepatic glutamate production due to endocrine alterations in the fetoplacental unit. The authors concluded that placental AA metabolism is altered in GDM pregnancies [26]. Similarly, our study observed a significant decrease in maternal serum glutamine levels in GDM cases, with a particularly sharp reduction in the subgroup of GDM pregnancies with normal-weight fetuses. This finding may indicate a compensatory role of glutamine in fetal weight regulation.

Another noteworthy finding was the elevated serine concentration in umbilical cord blood and amniotic fluid, which correlated with fetal weight (appropriate for gestational age vs. macrosomia). Several studies assessing metabolic disturbances in GDM have reported altered concentrations of amino acids such as alanine, valine, and serine [27,28,29]. In our study, GDM patients exhibited increased valine levels in AF and maternal venous blood, with subgroup analysis revealing this elevation exclusively in AF of GDM cases with fetal macrosomia. These amino acids may contribute to blood glucose modulation by influencing cellular energy metabolism, thereby potentially affecting transplacental nutrient transfer and fetal weight regulation [30].

Thus, a comparative analysis of currently available data and the original results obtained in this study reveals significant heterogeneity, likely influenced by sample size, ethnic variations, and other environmental factors that substantially affect the selection of metabolomic biomarkers for GDM. To validate these findings, large-scale multicenter studies should be conducted, accounting for the “phenotypic characteristics” of different geographic regions and establishing region-specific reference values for amino acid profiles.

Despite the pilot nature of this study, the obtained results are promising and open new avenues for developing an amino-acid-based biomarker panel for early GDM diagnosis and macrosomia prediction, as well as personalized metabolic correction in GDM patients, including dietary interventions tailored to amino acid metabolism alterations. These findings also hold significant potential for further investigation into GDM pathogenesis, particularly regarding placental amino acid transport and the role of epigenetic factors.

## 4. Materials and Methods

### 4.1. Study Design and Sample Collection

The study material was collected at the I.D. Evtushenko Regional Perinatal Center (Tomsk, Russia) between January 2024 and March 2025.

From an initial cohort of 2000 women undergoing first-trimester prenatal screening (11–13.6 weeks) at the Tomsk Perinatal Center, we prospectively selected 94 mother–newborn dyads for postpartum analysis, stratifying them into Group I (GDM-affected, n = 53) and Group II (non-GDM controls, n = 41); subsequent stratification created three clinically relevant subgroups: Ia (GDM with macrosomia, ≥4000 g, n = 23), Ib (GDM with normal birthweight, 2501–3999 g, n = 30), and II (non-GDM controls, 2501–3999 g, n = 36 after excluding 5 cases exceeding 3999 g), in this ethically approved (Kulakov National Medical Research Center IRB Protocol #4, 04/18/2024) multicenter collaboration between Siberian State Medical University (Tomsk) and Kulakov Center (Moscow), with all participants providing written informed consent prior to enrollment.

**Inclusion Criteria:** For the main group (GDM): Caucasian ethnicity, singleton pregnancy, neonatal birth weight 2501–4999 g with confirmed GDM diagnosis, and written informed consent. For the control group (non-GDM): Caucasian ethnicity, absence of GDM, singleton pregnancy, neonatal birth weight 2501–4999 g, and written informed consent. All participants were required to: (1) undergo comprehensive prenatal evaluation including three ultrasound screenings with venous blood sampling, (2) complete oral glucose tolerance testing at 24–28 weeks gestation, and (3) deliver at the I.D. Evtushenko Regional Perinatal Center (Tomsk).

**Exclusion Criteria:** Type 1 or 2 diabetes mellitus; any decompensated somatic pathology; oncological diseases; autoimmune disorders; bronchial asthma requiring medication-controlled management; and multifetal pregnancies.

The frequency of insulin therapy administration in women with GDM was comparable between fetal macrosomia (54.2%) and normosomia (59.1%) groups. During the first and third trimesters, only 2 patients with macrosomic fetuses and 2 with normosomic fetuses required insulin therapy initiation, while the second trimester saw insulin requirement in just 4 normosomic cases. In two study participants, GDM was diagnosed during the first trimester based on biochemical blood analysis (fasting glucose levels exceeding threshold values) and endocrinologist evaluation. Given their rapidly progressing insulin resistance and early-onset GDM, insulin therapy was initiated in the first trimester. These patients were retained in the study because their GDM diagnosis was unequivocally confirmed and properly documented in medical records, despite not undergoing oral glucose tolerance testing. Dietary modification with elimination of rapidly absorbable carbohydrates served as the primary therapeutic approach for hyperglycemia correction across all study participants.

All pregnant women underwent standardized ultrasound examinations at established gestational windows (11–14 weeks, 18–21 weeks, and 30–32 weeks). GDM diagnosis was confirmed through a 75 g oral glucose tolerance test (OGTT) performed after 8–14 h of overnight fasting. Venous plasma glucose levels were measured within 30 min of initial blood collection, with test termination if fasting glucose exceeded 5.1 mmol/L. For continuing tests, participants consumed within 5 min a glucose solution (LLC “Glukofan” Dolgoprudny, Moscow Region, Russia) containing 75 g anhydrous glucose dissolved in 250–300 mL of warm (37–40 °C) non-carbonated water. Subsequent venous plasma glucose measurements were obtained at 1 h and 2 h intervals post glucose load. Diagnostic thresholds for GDM and overt diabetes mellitus are detailed in Table 3 and Table 4.

Maternal blood and umbilical cord blood samples were collected using a vacuum system into sterile 9 mL S-Monovette tubes (Sarstedt AG and Co KG, Nümbrecht, Germany) containing clot activator and separation gel, following a 12 h fasting period. The serum was subsequently centrifuged at 700 g for 10 min at 4 °C, after which the supernatant was carefully aspirated using a specialized pipette and transferred into sterile Eppendorf tubes (Eppendorf, Hamburg, Germany) (1 mL aliquots). These aliquots were immediately frozen and stored at −80 °C until analysis. Amniotic fluid was collected into 2 mL plastic tubes, centrifuged at 700 g for 10 min at 4 °C, and 1 mL of the supernatant was transferred and stored at −80 °C until analysis. For all participants, maternal blood collection was performed during the designated study period between 37.0 and 40.6 weeks of gestation. Umbilical cord blood and amniotic fluid were collected during delivery. Prior to HPLC-MS/MS analysis, all blood specimens were maintained at −80 °C in the Biobank facility of Siberian State Medical University (Tomsk) to ensure sample integrity and stability.

### 4.2. Sample Preparation

To 50 μL of each sample, we added 50 μL of internal standard mixture solution (1.4 μg/mL per standard), vortex-mixed for 1 min, followed by sequential addition of 100 μL purified water (Sigma-Aldrich, St. Louis, MO, USA) and 500 μL chloroform/methanol (Sigma-Aldrich, St. Louis, MO, USA) (2:1, *v*/*v*). After 10 min of vortex mixing, samples were centrifuged at 15,000× *g* for 10 min. We then transferred 50 μL of the upper aqueous-methanolic layer to a clean microcentrifuge tube (Eppendorf, Hamburg, Germany) and evaporated it under a nitrogen stream at 60 °C. The residue was derivatized with 200 μL 3N HCl in 2-butanol (Sigma-Aldrich, St. Louis, MO, USA) at 60 °C for 15 min, followed by nitrogen evaporation at 60 °C. The dried extract was reconstituted in 300 μL methanol/water (Sigma-Aldrich, St. Louis, MO, USA) (1:1, *v*/*v*), vortex-mixed for 10 min, centrifuged at 15,000× *g* for 10 min, and 150 μL supernatant was transferred to an HPLC vial (Waters, Milford, MA, USA) with insert. Chromatographic analysis was performed with 1 μL injection volume.

### 4.3. LC-MS/MS Analysis

Amino acid profiling was performed using high-performance liquid chromatography coupled with tandem mass spectrometry (HPLC-MS/MS) on an analytical system comprising an Acquity I-Class liquid chromatograph (Waters, USA) and a Xevo TQ-XS triple quadrupole mass spectrometer (Waters, Milford, MA, USA) equipped with an electrospray ionization source. Chromatographic separation was achieved using an Agilent ZORBAX Eclipse XDB-C18 column (Agilent, Santa Clara, CA, USA) (2.1 mm× 100 mm, 1.8 μm particle size) maintained at 30 °C with a constant flow rate of 250 μL/min. The mobile phase consisted of (A) 10 mM ammonium acetate in water and (B) 0.1% formic acid in acetonitrile, with the following gradient elution program: 100% A (0–0.1 min), linear decrease from 95% to 75% A (0.1–19.0 min), 5% A (19.1–21.0 min), and re-equilibration at 100% A (21.1–30 min). The total chromatographic run time was 30 min.

The list of analyzed amino acids and their corresponding internal standards is provided in Appendix A. Calibration curves for all amino acids were generated using 10 amino acid concentration levels and 3 concentration levels were used for the analysis quality control (Appendix A). All measured quality control concentrations were within 15% of reference values.

### 4.4. Statistical Analysis

To assess the association between amino acid levels and GDM, we calculated the mutual information (MI) index [31], considering amino acids with MI values ≥ 50% of the maximum observed MI as potential markers. Candidate markers were further evaluated using Mann–Whitney U tests (with *p* < 0.05 considered significant) and median ratios, with final selection requiring either (1) significant Mann–Whitney results plus median ratios ≠ 1 or (2) absolute binary logarithms of median ratios > 1. For analyzing GDM and fetal macrosomia associations, we computed Shapley values [32] using a random forest classifier [33], designating amino acids as markers when their mean absolute Shapley values exceeded 50% of the maximum observed mean absolute Shapley value. Also, Kruskal–Wallis tests with Dunn tests were performed.

Marker enrichment for discriminating (1) control/GDM groups and (2) the “GDM+macrosomia” group from other metabolic pathways was evaluated using ConsensusPathDB [34], with significance threshold set at uncorrected *p* < 0.01. The top 10 most significant pathways were selected based on maximum geometric mean values (calculated as: [number of pathway-included markers]/[total pathway metabolites] and minimal false discovery rates (−log10 (FDR).

We performed Spearman’s correlation analysis to assess relationships between amino acid levels and clinical parameters in women with gestational diabetes mellitus (GDM), with statistical significance set at *p* < 0.05. To evaluate the association of clinical parameters with both GDM development and neonatal birth weight, we computed Shapley values [32] using a random forest classifier [33]. Potential biomarkers were identified as parameters with mean absolute Shapley values ≥ 50% of the maximum observed mean absolute Shapley value. All statistical analyses were conducted using custom R scripts (version 4.3.1, Viene, Austria) [35], leveraging the following packages: praznik (v11.0.0, Warsaw, Poland) [36] for feature selection, ggplot2 (v3.5.1, Posit Software, Boston, Massachusetts, USA) [37] and reshape2 (v1.4.4, Posit Software, Boston, Massachusetts, USA) [38] for data visualization, ranger (v0.17.0, Bremen, Germany) [39] for random forest modeling, and kernelshap (v0.7.0, ETH Zurich, Zurich, Switzerland) [40] for Shapley value computation.

## 5. Conclusions

This study presents a novel comprehensive analysis of the maternal–fetal amino acid profile system, uniquely examining all three biological compartments (maternal blood, umbilical cord blood, and amniotic fluid) in a large patient cohort. Our findings provide new insights into metabolic dysregulation in gestational diabetes mellitus and lay the groundwork for developing innovative diagnostic and preventive strategies for pregnancy complications.

The study revealed distinct amino acid profile alterations in the maternal–fetal system during GDM: decreased maternal serum levels of histidine, glutamine, and asparagine suggest impaired placental transport and increased fetal demand, while elevated amniotic fluid concentrations of leucine, isoleucine, serine, and threonine, along with increased cord blood isoleucine/serine and decreased arginine, indicate disrupted fetal anabolic processes potentially contributing to macrosomia development. Notably, 5-OH-lysine emerged as a GDM-specific biomarker independent of fetal weight, providing new insights into the metabolic pathophysiology of diabetic pregnancies and highlighting potential targets for diagnostic and therapeutic interventions.

Our findings align with previous reports demonstrating consistent alterations in amino acid concentrations (alanine, valine, serine, glutamine, and arginine) associated with GDM and fetal macrosomia. However, significant variability across studies—likely attributable to methodological, demographic, and analytical differences—underscores the current challenges in establishing reliable metabolomic biomarkers for GDM and macrosomia. This heterogeneity highlights the critical need for large-scale, multicenter validation studies to standardize biomarker identification and clinical translation.

## Figures and Tables

**Figure 1 ijms-26-08351-f001:**
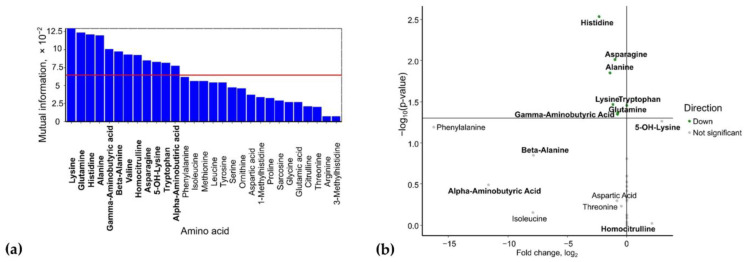
(**a**) Mutual information scores of serum amino acid levels. (**b**) Volcano plot of maternal serum amino acid levels in GDM. Amino acids identified as potential biomarkers based on mutual information index values are highlighted in bold.

**Figure 2 ijms-26-08351-f002:**
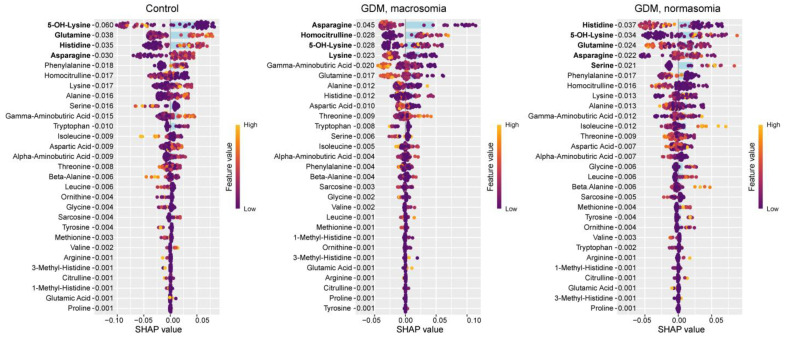
SHAP values of maternal serum amino acids for discrimination between the control group, GDM with macrosomia subgroup, and GDM with normosomia subgroup. Amino acids potentially serving as group biomarkers are highlighted in bold. Numeric labels adjacent to amino acids indicate the mean absolute SHAP values. Amino acids potentially serving as group biomarkers are highlighted in bold.

**Figure 3 ijms-26-08351-f003:**
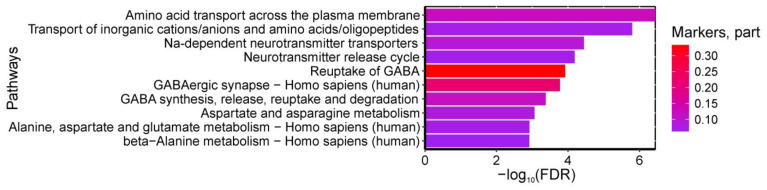
Most significantly enriched serum metabolic pathways discriminating GDM from controls, showing false discovery probability (−log10). Color intensity represents the proportion of pathway metabolites identified as GDM biomarkers.

**Figure 4 ijms-26-08351-f004:**
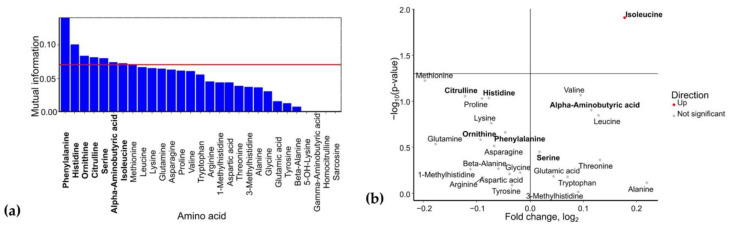
(**a**) Mutual information scores of amino acid levels in umbilical cord blood for GDM prediction. (**b**) Volcano plot of amino acid levels in umbilical cord blood from GDM cases. Amino acids identified as potential biomarkers based on mutual information index values are highlighted in bold.

**Figure 5 ijms-26-08351-f005:**
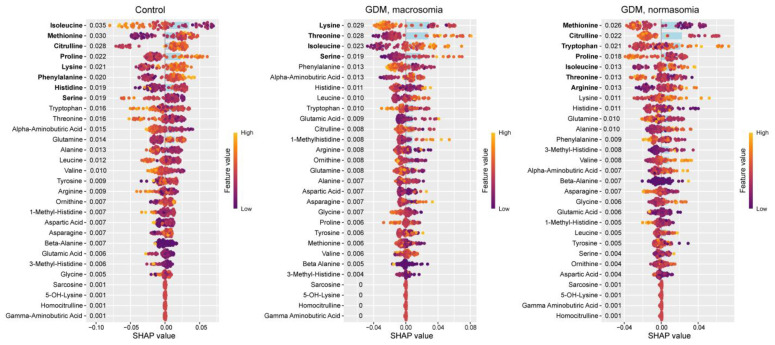
SHAP values of umbilical cord blood amino acids for discrimination between the control group, GDM with macrosomia subgroup, and GDM with normosomia subgroup. Amino acids potentially serving as group biomarkers are highlighted in bold. Numeric labels indicate mean absolute SHAP values.Amino acids potentially serving as group biomarkers are highlighted in bold.

**Figure 6 ijms-26-08351-f006:**
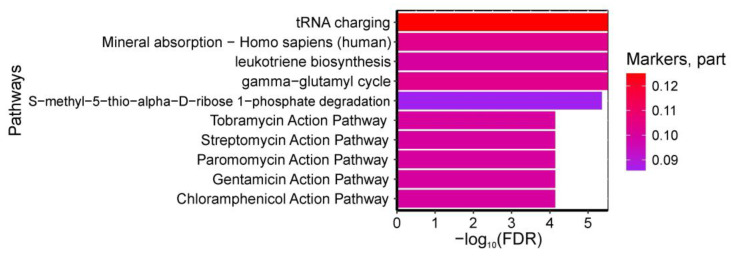
Most significantly enriched pathways associated with GDM+macrosomia biomarkers in umbilical cord blood, showing false discovery probability (−log10). Color intensity reflects the proportion of metabolites within each pathway identified as GDM+macrosomia markers.

**Figure 7 ijms-26-08351-f007:**
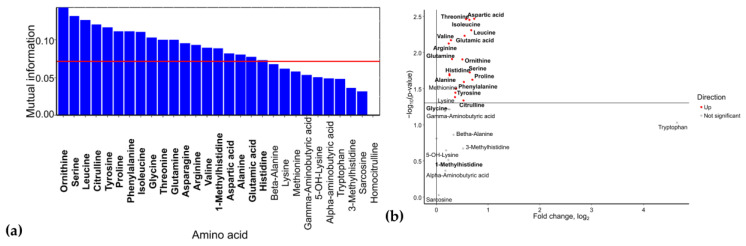
(**a**) Mutual information scores of amniotic fluid amino acid levels for GDM prediction. (**b**) Volcano plot of amniotic fluid amino acid levels in GDM cases. Amino acids identified as potential biomarkers based on mutual information index values are highlighted in bold.

**Figure 8 ijms-26-08351-f008:**
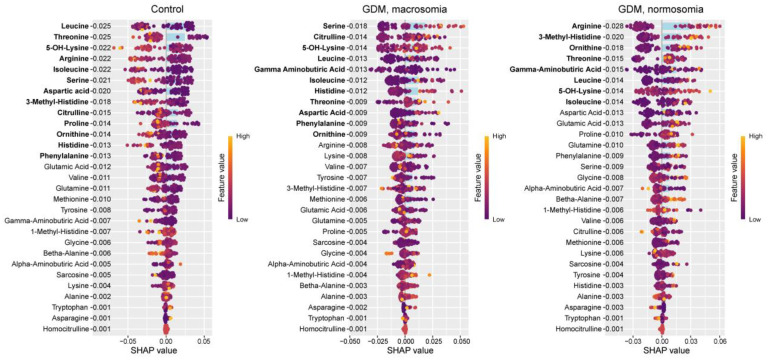
SHAP values of amniotic fluid amino acids for discrimination between the control group, GDM with macrosomia subgroup, and GDM with normosomia subgroup. Potential group-specific biomarker amino acids are highlighted in bold, with numeric labels indicating mean absolute SHAP values.

**Figure 9 ijms-26-08351-f009:**
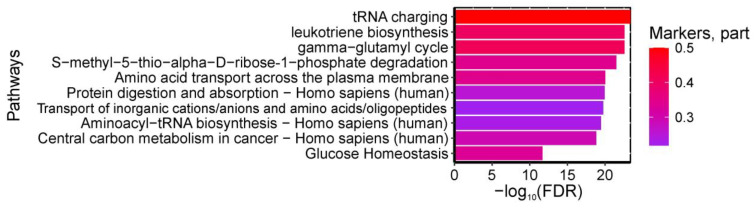
Most significantly enriched metabolic pathways based on amniotic fluid biomarkers for GDM/control discrimination, showing false discovery probability (−log10). Color intensity represents the proportion of pathway metabolites identified as GDM markers.

**Figure 10 ijms-26-08351-f010:**
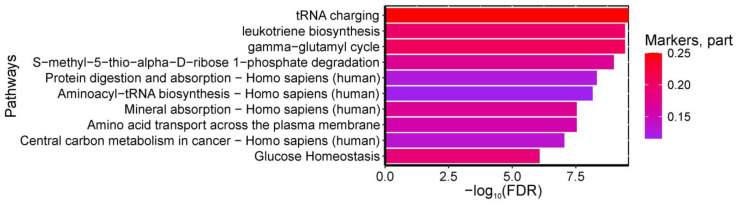
Most significantly enriched metabolic pathways based on amniotic fluid biomarkers for GDM+macrosomia discrimination, showing false discovery probability (−log10). Color gradient represents the proportion of pathway metabolites identified as GDM markers, with increasing intensity indicating higher biomarker representation.

**Figure 11 ijms-26-08351-f011:**
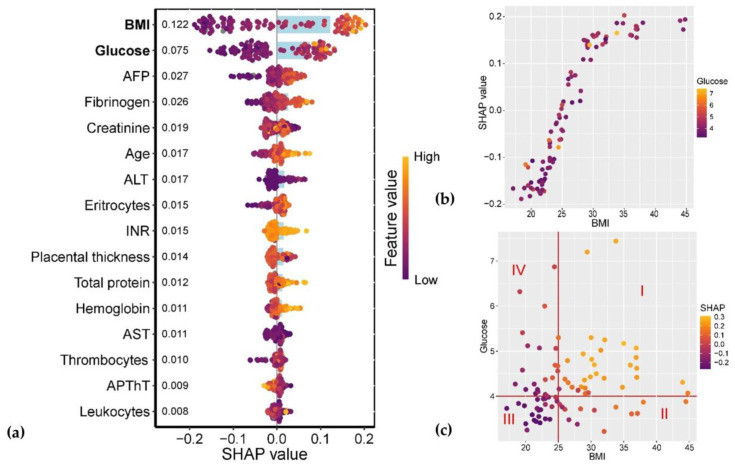
(**a**) SHAP values of clinical parameters for GDM association analysis. Parameters with the most significant associations are highlighted in bold, with numeric labels indicating mean absolute SHAP values. (**b**) Relationship between BMI levels and their corresponding SHAP values. (**c**) Combined association of BMI and glucose levels with their aggregated SHAP values.

**Figure 12 ijms-26-08351-f012:**
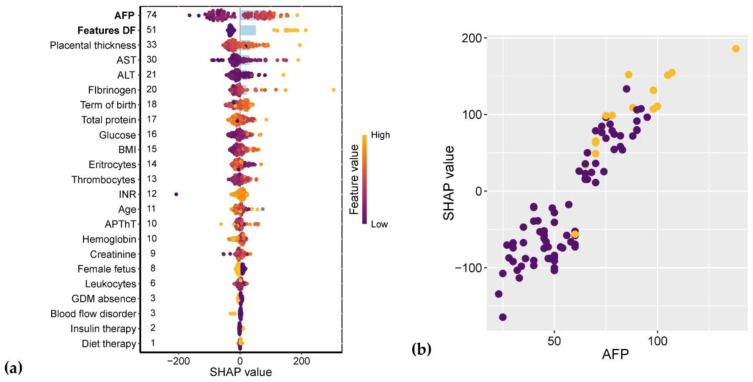
(**a**) SHAP values of clinical parameters for their association with neonatal birth weight. Parameters demonstrating the most significant associations are highlighted in bold, with numerical labels indicating mean absolute SHAP values. (**b**) Relationship between amniotic fluid pocket depth (AFP) and corresponding SHAP values. Patients without ultrasound markers of diabetic fetopathy (DF) are shown in purple, while those with DF diagnosis are marked in yellow.

**Figure 13 ijms-26-08351-f013:**
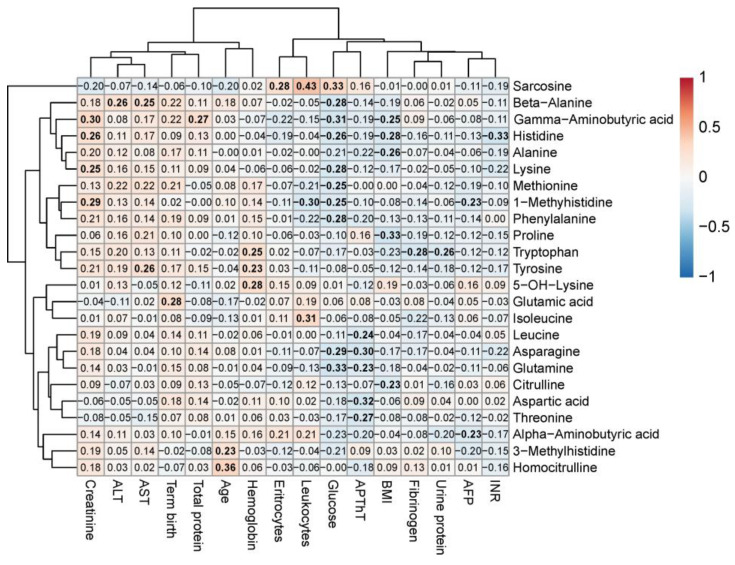
Correlation matrix of serum amino acids and clinical parameters with at least one statistically significant association. Amino acids and parameters are ordered by similarity of correlation coefficients. Statistically significant correlations are highlighted in bold.

**Table 1 ijms-26-08351-t001:** Discrimination power of potential serum markers for discrimination of control and GDM groups and normosomia and macrosomia GDM subgroups: area under receiver operating curves (AUC), optimal sensitivity and specificity.

	Control/GDM			(GDM) Normosomia/Macrosomia		
	AUC	Sensitivity	Specificity	AUC	Sensitivity	Specificity
Glutamine	0.64	0.81	0.55	0.60	0.38	0.90
Histidine	0.70	0.84	0.59	0.48	0.50	0.74
Asparagine	0.68	0.67	0.68	0.71	0.63	0.84

**Table 2 ijms-26-08351-t002:** Comparative Amino Acid Profiling in GDM Across Biological Compartments. ↓ is decreasing level, ↑ is increasing level.

Amino Acid	Maternal Venous Blood	Cord Blood	Amniotic Fluid	*p*-Value	Major Changes in GDM	Association with Macrosomia
Histidine	↓	↓	↓	*p* = 0.003, *p* * = 0.09, *p* ** = 0.02	Associated with GDM Decrease in maternal and cord blood, increase in AF	Non-specific
Glutamine	↓	↓	↑	*p* = 0.04, *p* * = 0.29, *p* ** = 0.01	Associated with GDM Decrease in maternal and umbilical cord blood, increase in AF	Non-specific
5-OH-Lysine	↑	–	↑	*p* = 0.05, *p* * = 1, *p* ** = 0.2	Associated with GDM Increase in maternal blood and AF	Non-specific
Asparagine	↓	-	-	*p* = 0.01, *p* * = 0.30, *p* ** = 0.16	Decreased in maternal blood	Associated with macrosomia
Isoleucine	–	↑	↑	*p* = 0.7, *p* * = 0.01, *p* ** = 0.004	Increase in umbilical cord blood and AF	Associated with macrosomia
Serine	–	↑	↑	*p* = 0.75, *p* * = 0.35, *p* ** = 0.02	Increase in umbilical cord blood and AF	Associated with macrosomia
Threonine	–	↑	↑	*p* = 0.59, *p* * = 0.43, *p* ** = 0.003	Increase in umbilical cord blood and AF	Associated with macrosomia
Arginine	–	↓	↑	*p* = 0.82, *p* * = 0.67, *p* ** = 0.007	Decreased in cord blood, increased in AF	Associated with macrosomia

*p* without *—comparing serum venous blood, *p* *—comparing serum cord blood, *p* **—comparing amniotic fluid. AF—amniotic fluid.

**Table 3 ijms-26-08351-t003:** Diagnostic Thresholds for GDM Using the 75 g Oral Glucose Tolerance Test.

Venous Plasma Glucose	mmol/L	mg/dL
Fasting	≥5.1	≥92
1 h postload	≥10.0	≥180
2 h postload	≥8.5	≥153

**Table 4 ijms-26-08351-t004:** Diagnostic Thresholds for Overt Diabetes Mellitus in Pregnancy.

Fasting Venous Plasma Glucose	≥7.0 mmol/L	≥126 mg/dL
Glycated hemoglobin (HbA1c)	≥6.5%	
Random venous plasma glucose	≥11.1 mmol/L	≥200 mg/dL

## Data Availability

Data are contained within the article and Appendix A.

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
