# Peer review of "Amino Acid Profile Alterations in the Mother–Fetus System in Gestational Diabetes Mellitus and Macrosomia"

_ijms, 2025, doi:10.3390/ijms26178351_

Round 1
Reviewer 1 Report
Comments and Suggestions for Authors
This study aims to compare the amino acid variation profiles among three groups: non-GDM, GDM with macrosomia, and GDM with normosomia. Free amino acid profiles from three biological sample types—maternal serum, umbilical cord blood, and amniotic fluid—were analyzed using LC-MS/MS, followed by statistical analysis and machine learning approaches to identify group-specific alterations. The results reveal several key amino acid markers associated with GDM, which may hold potential for future diagnostic applications. However, several aspects require clarification before the study is suitable for publication:
- The manuscript presents alterations in free amino acid profiles and their associated metabolic pathways across the sample groups. To ensure scientific rigor, the authors should clarify whether the detected amino acids are exclusively free-form, and not products of protein degradation.
- Regarding the false discovery rate (FDR) method employed to identify significant metabolic pathways, further detail is needed on the cut-off or fitting values used. Current reporting lacks specificity and includes some seemingly unrelated pathways—e.g., antibiotic-related pathways identified in umbilical cord blood—raising questions about biological plausibility.
- The LC-MS/MS analyses were performed on samples collected between 37 and 40.6 weeks of gestation, from patients already diagnosed and categorized into the respective study groups. Therefore, the authors should reconsider the claim that these findings could contribute to early diagnostic marker development for GDM with macrosomia, as the dataset does not represent early-stage disease conditions.
Minor Revisions:
- All manuscript content should be presented in English. Please revise labeling in Figure 13 and the concentration units in Table 3, and in Page 20, Line 596 accordingly.
- In Table 1 (Page 15, Line 374) and again on Page 20, Line 595, the definition of P** is repeated while the explanation for P* is missing. Please address this inconsistency.
- The title of Table S1 in the supplementary file should be specified for clarity and proper referencing.
Author Response
Thank you to the reviewer for thoroughly reviewing our article
This study aims to compare the amino acid variation profiles among three groups: non-GDM, GDM with macrosomia, and GDM with normosomia. Free amino acid profiles from three biological sample types—maternal serum, umbilical cord blood, and amniotic fluid—were analyzed using LC-MS/MS, followed by statistical analysis and machine learning approaches to identify group-specific alterations. The results reveal several key amino acid markers associated with GDM, which may hold potential for future diagnostic applications. However, several aspects require clarification before the study is suitable for publication:
The manuscript presents alterations in free amino acid profiles and their associated metabolic pathways across the sample groups. To ensure scientific rigor, the authors should clarify whether the detected amino acids are exclusively free-form, and not products of protein degradation.
Answer: Free amino acids were analyzed in this study. Minimal or no protein degradation was assumed during sample collection and preparation. All samples were stored at -80°C and proteins were denatured during sample preparation, so enzymatic protein digestion activity is not assumed.
Regarding the false discovery rate (FDR) method employed to identify significant metabolic pathways, further detail is needed on the cut-off or fitting values used. Current reporting lacks specificity and includes some seemingly unrelated pathways—e.g., antibiotic-related pathways identified in umbilical cord blood—raising questions about biological plausibility.
Answer: At the beginning, unadjusted p-value were used for thresholding during pathway enrichment analysis, but in the case of the pathways shown in Figures 3, 6, 9, 10, the false discovery rate (adjusted p-value) <0.01. 0.01 is a usual threshold value. Information about this condition was added in the articles. Including isoluecine and threonine in antibiotics-related pathways was caused by fact, that threonine is isoleucine’s precursor and isoleucine by influence of aminoglycoside antibiotics missing included to aminoacid sequence, breaking the structure of the future protein. (10.1054/mehy.2001.1450, 10.1126/science.150.3701.1290)
The LC-MS/MS analyses were performed on samples collected between 37 and 40.6 weeks of gestation, from patients already diagnosed and categorized into the respective study groups. Therefore, the authors should reconsider the claim that these findings could contribute to early diagnostic marker development for GDM with macrosomia, as the dataset does not represent early-stage disease conditions.
Answer: The statement has been revised to: 'may contribute to the development of early diagnostic markers in the future, after validation in a large cohort of women in early pregnancy.'
Minor Revisions:
All manuscript content should be presented in English. Please revise labeling in Figure 13 and the concentration units in Table 3, and in Page 20, Line 596 accordingly.
Answer: corrected
In Table 1 (Page 15, Line 374) and again on Page 20, Line 595, the definition of P** is repeated while the explanation for P* is missing. Please address this inconsistency.
Answer: corrected
The title of Table S1 in the supplementary file should be specified for clarity and proper referencing.
Answer: corrected
Reviewer 2 Report
Comments and Suggestions for Authors
I thank the authors for the submitted article.
The strong point of the article is its systematic approach: for the first time, all three biological compartments (maternal blood, umbilical cord blood, and amniotic fluid) have been comprehensively evaluated, allowing for a deeper understanding of metabolic shifts in GDM. The results of the study are of great interest and can serve as a basis for improving the early diagnosis of GDM and macrosomia. However, some improvements are required in the presentation of the research results to enhance their clarity for readers.
The abstract is well-structured, but it requires a more specific statement of the purpose and revision of some details (see detailed comments below by line numbers).
Lines 13-17: The authors state the purpose as a search for biomarkers for prediction and diagnosis. However, the study design uses not only maternal blood samples but also umbilical cord blood, which is almost impossible to utilize for these purposes. The aim should be clarified and specified accordingly. The authors also refer to gestational diabetes mellitus in the introduction but do not mention fetal macrosomia, which appears later in the text.
Line 23: Please specify the exact values instead of general notation (e.g., |R| < 0.4, p < 0.05).
In the introduction, I recommend reducing the use of introductory words and phrases that do not add semantic value, such as “however”.
Lines 32, 35: Please specify the reference year for these data.
Line 34: Fetal macrosomia must be defined.
Line 41: This information should be provided earlier, together with the macrosomia definition (i.e., birth weight > 4,000 g).
Lines 45-50: These sentences are uninformative. The authors should specify the loci identified by GWAS, rather than using vague language. The same applies to methylation discussion.
It is also recommended to explain the relationship between genetics, epigenetics, and metabolomics, as discussed further on, otherwise the connection among these aspects remains unclear.
Line 55: “However, most published studies focus solely on maternal metabolic changes.” Please elaborate on the nature of these metabolic changes in 1–3 sentences.
Line 65: Please remove “However”.
Results section:
There is no clinical characterization of the analyzed groups in the results section. Please include this information in the results or as Supplementary files, and highlight the key differences between the groups at the beginning of the section.
Lines 84-87: Exact p-values should be indicated; if possible, provide fold changes as well.
Line 94: Please define “SHAP”.
Lines 97-98: Provide quantitative information regarding changes (e.g., how many times asparagine and lysine levels decrease, and homocitrulline and 5-OH-lysine levels increase). Apply this suggestion also to lines 100-101, 135-137, 146-150, 184-187, 200, 204, 207-219. Alternatively, highlight amino acids with the largest fold change.
Section 2.5, “Clinical markers”, along with the group characteristics, should be positioned at the beginning of the section. The clinical characteristics of the analyzed group and statistical analysis are currently missing; please add a table with this information.
It would be better to move Table 1 to the end of the results section, as a summary of the data obtained.
Discussion section
This part needs to be thoroughly revised. Most of the discussion reviews previous studies rather extensively rather than analyzing and comparing them with the current results.
The authors should address the specificity of the proposed biomarkers: are these changes unique to pregnancy-related pathologies, or might they also occur in other diseases?
Line 456: This subsection would be better titled “4.1 Study design and sample collection”.
Lines 543-544: The abbreviation “GDM” was previously introduced.
In the Supplementary Table 1: Please specify units of measurement. Add clarification for symbols “*”, “**”, and “***” beneath Supplementary Table 1.
Author Response
Thank you to the reviewer for thoroughly reviewing our article
I thank the authors for the submitted article.
The strong point of the article is its systematic approach: for the first time, all three biological compartments (maternal blood, umbilical cord blood, and amniotic fluid) have been comprehensively evaluated, allowing for a deeper understanding of metabolic shifts in GDM. The results of the study are of great interest and can serve as a basis for improving the early diagnosis of GDM and macrosomia. However, some improvements are required in the presentation of the research results to enhance their clarity for readers.
The abstract is well-structured, but it requires a more specific statement of the purpose and revision of some details (see detailed comments below by line numbers).
Lines 13-17: The authors state the purpose as a search for biomarkers for prediction and diagnosis. However, the study design uses not only maternal blood samples but also umbilical cord blood, which is almost impossible to utilize for these purposes. The aim should be clarified and specified accordingly. The authors also refer to gestational diabetes mellitus in the introduction but do not mention fetal macrosomia, which appears later in the text.
Answer: the text was corrected. Additional material describing the issue of macrosomia has been included: Gestational diabetes mellitus (GDM) and associated fetal macrosomia are becoming increasingly significant global health challenges, necessitating the development of new diagnostic and prognostic approaches.
Line 23: Please specify the exact values instead of general notation (e.g., |R| < 0.4, p < 0.05).
Answer: Corrected
In the introduction, I recommend reducing the use of introductory words and phrases that do not add semantic value, such as “however”.
Answer: Corrected
Lines 32, 35: Please specify the reference year for these data.
Answer: the year was specified
Line 34: Fetal macrosomia must be defined.
Answer: The text was added: According to WHO criteria, fetal macrosomia is defined as a condition where the estimated or actual fetal/neonatal body weight exceeds 4,000 grams (regardless of gestational age) (WHO guidelines on GDM, 2013)
Line 41: This information should be provided earlier, together with the macrosomia definition (i.e., birth weight > 4,000 g).
Answer: Corrected
Lines 45-50: These sentences are uninformative. The authors should specify the loci identified by GWAS, rather than using vague language. The same applies to methylation discussion.
It is also recommended to explain the relationship between genetics, epigenetics, and metabolomics, as discussed further on, otherwise the connection among these aspects remains unclear.
Answer: we believe additional explanations may overburden the text. This material is included to highlight the connection between amino acids and biological processes in the maternal-fetal system, drawing attention to this aspect of GDM. However, we do not explore it in detail, as this was not the primary focus of our study
Line 55: “However, most published studies focus solely on maternal metabolic changes.” Please elaborate on the nature of these metabolic changes in 1–3 sentences.
Answer: Detailed explanations regarding the pathogenesis of GDM and macrosomia are provided in the Discussion section. The statement 'However, most published studies focus exclusively on maternal metabolic alterations' implies - within the context of this paragraph - that the majority of research to date (as represented in the international PubMed database) has investigated only maternal blood in GDM. In contrast, our study analyzed maternal blood, umbilical cord blood, and amniotic fluid in GDM cases
Line 65: Please remove “However”.
Answer: Corrected
Results section:
There is no clinical characterization of the analyzed groups in the results section. Please include this information in the results or as Supplementary files, and highlight the key differences between the groups at the beginning of the section.
Answer: The table S4 and appropriate text was added: The groups were comparable in age, gestational age at delivery, and fetal sex (Table S4). Mean maternal age was 33.5 years in the GDM group versus 34 years in controls. All participants delivered at 39 weeks of gestation (38; 39). Neonatal sex distribution showed no significant intergroup difference (p=0.49). Pre-pregnancy BMI was lower in controls (22 kg/m²). Recent evidence identifies pre-pregnancy BMI as a key clinical predictor of GDM development. Notably, women with BMI 22 kg/m² (21; 24) who had comparable total weight gain to the GDM-with-macrosomia subgroup (13 kg (11; 16) vs. 14 kg (11; 16), respectively) did not develop GDM. Of particular interest, the lowest weight gain occurred in the GDM subgroup with-out macrosomia (11 kg (9; 14), p<0.02), highlighting the significant impact of dietary con-trol and lifestyle modifications on preventing macrosomia in GDM. Cesarean delivery rates were significantly higher in GDM women versus controls (p=0.003), reaching 70% in the macrosomia subgroup. Ultrasound evaluation revealed significant increases in placental thickness and amniotic fluid pocket depth in GDM pa-tients with macrosomia (p=0.007 and p=0.03, respectively). Detailed clinical characteristics and routine laboratory/imaging data extracted from primary medical records are presented in Table S5.
Lines 84-87: Exact p-values should be indicated; if possible, provide fold changes as well.
Answer: Exact p-values and fold changes (FC) were added.
Line 94: Please define “SHAP”.
Answer: SHAP is SHapley Additive Explanation, measure of the effect to the result. The description has been added to the text.
Lines 97-98: Provide quantitative information regarding changes (e.g., how many times asparagine and lysine levels decrease, and homocitrulline and 5-OH-lysine levels increase). Apply this suggestion also to lines 100-101, 135-137, 146-150, 184-187, 200, 204, 207-219. Alternatively, highlight amino acids with the largest fold change.
Answer: Fold changes and p-values for the most impotant aminoacids in binary compatison were added. Fold changes for three-class comparison (control /GDM + macrosomia / GDM +normosomia) were added.
Section 2.5, “Clinical markers”, along with the group characteristics, should be positioned at the beginning of the section. The clinical characteristics of the analyzed group and statistical analysis are currently missing; please add a table with this information.
Answer: The table S5 was added.
It would be better to move Table 1 to the end of the results section, as a summary of the data obtained.
Answer: In our view, the table is more informative when placed within the main text. The text has been modified accordingly
Discussion section
This part needs to be thoroughly revised. Most of the discussion reviews previous studies rather extensively rather than analyzing and comparing them with the current results.
The authors should address the specificity of the proposed biomarkers: are these changes unique to pregnancy-related pathologies, or might they also occur in other diseases?
Answer: The design of this study does not include testing for specificity. For this purpose, other patient groups need to be formed
Line 456: This subsection would be better titled “4.1 Study design and sample collection”.
Answer: The title was modified
Lines 543-544: The abbreviation “GDM” was previously introduced.
Answer: Corrected
In the Supplementary Table 1: Please specify units of measurement. Add clarification for symbols “*”, “**”, and “***” beneath Supplementary Table 1.
Answer: Corrected: Units were added, clarification was added in the title.
Reviewer 3 Report
Comments and Suggestions for Authors
The manuscript analyzes amino acid levels in patients with GDM and in neonatal biological fluids. The observed alterations are tissue-specific, and the results obtained are of scientific relevance. However, there are several concerns that need to be addressed before the manuscript can be considered suitable for publication.
In several parts, the manuscript has not been translated from the original language, and numerous typos remain in the figures, making comprehension difficult (Figure 13 and supplemental).
In the sample collection section, there is no mention of how the amniotic fluid or cord blood were collected, processed, and stored. the same for the clinical measured parameters.
There is no information on maternal characteristics such as age, BMI, or gestational age. Additionally, data on neonatal birth weight and sex are completely missing. This appears to be of fundamental importance, as the observed differences could potentially be correlated with these parameters.
I have a concern that targeted LC-MS/MS for amino acids is appropriate and internal standards are mentioned, but no information on analytical QC samples, coefficient of variation or batch correction is provided, limiting confidence in small percentage changes.
lines 95-96 "These same amino acids were previously identified as potential biomarkers for distinguishing..." By whom were they identified?
The description of the results would benefit from a univariate comparison to better understand the actual differences between groups in each biological fluid analyzed. The separate analysis for each group, although of relevant interest, leaves room for misinterpretation. Please consider using ANOVA and post-hoc tests to better assess the differences between groups.
Author Response
Thank you to the reviewer for thoroughly reviewing our article
The manuscript analyzes amino acid levels in patients with GDM and in neonatal biological fluids. The observed alterations are tissue-specific, and the results obtained are of scientific relevance. However, there are several concerns that need to be addressed before the manuscript can be considered suitable for publication.
In several parts, the manuscript has not been translated from the original language, and numerous typos remain in the figures, making comprehension difficult (Figure 13 and supplemental).
Answer: The text was corrected
In the sample collection section, there is no mention of how the amniotic fluid or cord blood were collected, processed, and stored. the same for the clinical measured parameters.
Answer: Information on amniotic fluid and cord blood has been added to the Materials and Methods section.
There is no information on maternal characteristics such as age, BMI, or gestational age. Additionally, data on neonatal birth weight and sex are completely missing. This appears to be of fundamental importance, as the observed differences could potentially be correlated with these parameters.
Answer: Clinical data were added as Supplementary Table S5.
I have a concern that targeted LC-MS/MS for amino acids is appropriate and internal standards are mentioned, but no information on analytical QC samples, coefficient of variation or batch correction is provided, limiting confidence in small percentage changes.
Answer: Calibration curves for all amino acids were generated using 10 amino acid concentration levels and 3 concentration levels were used for the analysis quality control (Table S6). All measured quality control concentrations were within 15% of reference values.
lines 95-96 "These same amino acids were previously identified as potential biomarkers for distinguishing..." By whom were they identified?
Answer: the sentence was corrected.
The description of the results would benefit from a univariate comparison to better understand the actual differences between groups in each biological fluid analyzed. The separate analysis for each group, although of relevant interest, leaves room for misinterpretation. Please consider using ANOVA and post-hoc tests to better assess the differences between groups.
Answer: We created new supplementary tables with results of comparison three groups by Kruskall-Wallis test with Dunn’s test (Tables S1 – S3).
Reviewer 4 Report
Comments and Suggestions for Authors
This untargeted metabolomics study reported altered amino acid profiles in the maternal-fetal system of GDM and fetal macrosomia, with a well-designed and well-defined methodology. The results are innovative and have potential clinical valuable.
Comments
- Specificity and sensitivity of biomarkers: The article mentions glutamine, histidine, and asparagine as “potential” biomarkers, but does not provide preliminary data or analyses of their diagnostic performance (e.g., area under the ROC curve AUC, sensitivity, specificity). This information is essential to assess their actual value as early predictive markers.
- Mechanisms not well explored:The article mentions that machine learning and metabolic pathway analysis have revealed key metabolic pathways, but fewer have been linked to the pathophysiology of GDM and the occurrence of macrosomia. Adding speculation on the underlying mechanisms would be more compelling.
- Matching of controls: Whether the GDM and control groups were matched or statistically controlled for baseline characteristics such as age, gestational week, and BMI. This can affect the confidence of the results. It is recommended todetail in the main text.
- Limitations:The article ends with the phrase “standardize biomarker identification and clinical translation”. A more specific reference to possible confounding factors in this study (e.g., diet, medication use, comorbidities, etc.) and how the study controlled for these confounding factors?
Author Response
Thank you to the reviewer for thoroughly reviewing our article
This untargeted metabolomics study reported altered amino acid profiles in the maternal-fetal system of GDM and fetal macrosomia, with a well-designed and well-defined methodology. The results are innovative and have potential clinical valuable.
Comments
Specificity and sensitivity of biomarkers: The article mentions glutamine, histidine, and asparagine as “potential” biomarkers, but does not provide preliminary data or analyses of their diagnostic performance (e.g., area under the ROC curve AUC, sensitivity, specificity). This information is essential to assess their actual value as early predictive markers.
Answer: We added information abour operation curve analysis of the potential biomarkers for discrimination control and GDM and differentiation normosomia and macrosomia GDM-subgroups.
Mechanisms not well explored:The article mentions that machine learning and metabolic pathway analysis have revealed key metabolic pathways, but fewer have been linked to the pathophysiology of GDM and the occurrence of macrosomia. Adding speculation on the underlying mechanisms would be more compelling.
Answer: The most enriched pathways earlier were identified as pathways, associated with type 1 or type 2 diabetes melitus. Also, the pathways, associated with aminoacids translocation and aminoacid sequences synthesis, can be associated with macrosomia.
Matching of controls: Whether the GDM and control groups were matched or statistically controlled for baseline characteristics such as age, gestational week, and BMI. This can affect the confidence of the results. It is recommended todetail in the main text.
Answer: Age, gestational week and sex fetus are matched in our group.
Limitations:The article ends with the phrase “standardize biomarker identification and clinical translation”. A more specific reference to possible confounding factors in this study (e.g., diet, medication use, comorbidities, etc.) and how the study controlled for these confounding factors?
Answer: The primary limitations of our study were the relatively small sample size and potential regional specificities (Siberian population). We plan to address these limitations in future research by expanding our cohort through a multicenter study encompassing several regions, which should yield more precise data on the maternal-fetal metabolome in GDM. To minimize confounding factors and maximize data accuracy, we rigorously adhered to: Strict inclusion/exclusion criteria (selecting a homogeneous cohort with similar lifestyles, without special dietary regimens, medications, or severe comorbidities); Standardized protocols for biological sample collection, storage, and analysis.
Reviewer 5 Report
Comments and Suggestions for Authors
Gestational diabetes (GDM) is one of the most common metabolic complications occurring during pregnancy. According to global data, GDM affects about 14% of pregnant women worldwide. It is associated with an increased risk of complications during pregnancy for both the mother and the fetus. Possible complications include macrosomia, shoulder dystocia, fetal hypoglycemia, and increased caesarean section rate.
In this study, the authors analyzed amino acid profiles in the mother-fetus system to identify biomarkers linked to GDM and fetal macrosomia. They analyzed serum from maternal venous and umbilical cord blood, along with amniotic fluid.
The study revealed amino acid profile alterations in the maternal-fetal system during GDM: decreased maternal serum levels of histidine, glutamine and asparagine suggest impaired placental transport and increased fetal demand, while elevated amniotic fluid concentrations of leucine, isoleucine, serine and threonine, along with increased cord blood isoleucine/serine and decreased arginine, indicate disrupted fetal anabolic processes potentially contributing to macrosomia development, one of the most common complication of gestational diabetes mellitus.
This study is technically very well-performed, fairy well written and the findings are very interesting and informative. The analysis and discussion of the research deserves great recognition. However, one issue should be clarified.Line 481-484: “During the first and third trimesters, only 2 patients with macrosomic fetuses and 2 with normosomic fetuses required insulin therapy initiation, while the second trimester saw insulin requirement in just 4 normosomia (it should be change into normosomic) cases.
Why was insulin therapy conducted in the first trimester of pregnancy, since the diagnosis of GDM was made based on a test at 24-28 weeks of pregnancy, and patients with type I and II diabetes were excluded from the study? This should be explained.
In my opinion the manuscript is worth for publication.
Author Response
Thank you to the reviewer for thoroughly reviewing our article
Gestational diabetes (GDM) is one of the most common metabolic complications occurring during pregnancy. According to global data, GDM affects about 14% of pregnant women worldwide. It is associated with an increased risk of complications during pregnancy for both the mother and the fetus. Possible complications include macrosomia, shoulder dystocia, fetal hypoglycemia, and increased caesarean section rate.
In this study, the authors analyzed amino acid profiles in the mother-fetus system to identify biomarkers linked to GDM and fetal macrosomia. They analyzed serum from maternal venous and umbilical cord blood, along with amniotic fluid.
The study revealed amino acid profile alterations in the maternal-fetal system during GDM: decreased maternal serum levels of histidine, glutamine and asparagine suggest impaired placental transport and increased fetal demand, while elevated amniotic fluid concentrations of leucine, isoleucine, serine and threonine, along with increased cord blood isoleucine/serine and decreased arginine, indicate disrupted fetal anabolic processes potentially contributing to macrosomia development, one of the most common complication of gestational diabetes mellitus.
This study is technically very well-performed, fairy well written and the findings are very interesting and informative. The analysis and discussion of the research deserves great recognition. However, one issue should be clarified.Line 481-484: “During the first and third trimesters, only 2 patients with macrosomic fetuses and 2 with normosomic fetuses required insulin therapy initiation, while the second trimester saw insulin requirement in just 4 normosomia (it should be change into normosomic) cases.
Why was insulin therapy conducted in the first trimester of pregnancy, since the diagnosis of GDM was made based on a test at 24-28 weeks of pregnancy, and patients with type I and II diabetes were excluded from the study? This should be explained.
Answer: In two study participants, GDM was diagnosed during the first trimester based on biochemical blood analysis (fasting glucose levels exceeding threshold values) and endocrinologist evaluation. Given their rapidly progressing insulin resistance and early-onset GDM, insulin therapy was initiated in the first trimester. These patients were retained in the study because their GDM diagnosis was unequivocally confirmed and properly documented in medical records, despite not undergoing oral glucose tolerance testing. Text was added.
In my opinion the manuscript is worth for publication.
Round 2
Reviewer 2 Report
Comments and Suggestions for Authors
Thanks for the changes in the article.
Lines 45-50: If your research is not focused on genetics and epigenetics, their mention in the introduction is superfluous, unless you associate them with changes in the amino acid profile. Either eliminate this part, or logically link genetics, epigenetics, and changes in the amino acid profile.
I recommend that the authors add a brief description of the analyzed sample at the beginning of the results (sample size by group, age, BMI, gestation period) at the beginning of the "Results" section. This will provide the reader with a better understanding of the source data, while a detailed analysis of the links to the metabolomic profiles can be left in section 2.4 ("Clinical markers").
Author Response
We thank the reviewer for their thorough work in reviewing our manuscript
Lines 45-50: If your research is not focused on genetics and epigenetics, their mention in the introduction is superfluous, unless you associate them with changes in the amino acid profile. Either eliminate this part, or logically link genetics, epigenetics, and changes in the amino acid profile.
Answer: We have eliminated this part.
I recommend that the authors add a brief description of the analyzed sample at the beginning of the results (sample size by group, age, BMI, gestation period) at the beginning of the "Results" section. This will provide the reader with a better understanding of the source data, while a detailed analysis of the links to the metabolomic profiles can be left in section 2.4 ("Clinical markers").
Answer: We have added a brief description into the manuscript: “At the initial stage of the study, a prospective selection was made from a cohort of 2000 women who underwent first-trimester prenatal screening (11–13.6 weeks) at the Tomsk Perinatal Center. This selection resulted in 94 mother-newborn pairs, who were divided into two main groups: Group I (patients with GDM, n=53) and control Group II (non-GDM control group, n=41).
Subsequent stratification identified three clinically significant subgroups: Ia (GDM with macrosomia ≥4000 g, n=23), Ib (GDM with normal birth weight 2501–3999 g, n=30), and II (non-GDM control group, 2501–3999 g, n=36 after the exclusion of 5 cases exceeding 3999 g).”
Reviewer 3 Report
Comments and Suggestions for Authors
I thank the authors for having responded thoroughly to my requests. However, some concerns remain regarding the results.
- The authors reported that some newborns were delivered by caesarean section. Compared to spontaneous delivery, this may induce alterations or introduce a new variable. Are there any indications in this regard? Could anesthesia have modified and/or influenced serum amino acid levels?
- Approximately half of the patients were treated with insulin for GDM. Are there differences between treated and untreated patients in terms of blood parameters and measured variables?
- How does the difference in weight affect the observed differences? Body weight and BMI should also be included in the correlations to assess the dependence of the measured variables on these two parameters.
- Other minor revisions: in the tables, some decimals are indicated with a comma instead of a period. Please ensure consistency.
- In the discussion, it would be appropriate to add references regarding the studies cited in lines 400–402 and 425–430.
Author Response
We thank the reviewer for their thorough work in reviewing our manuscript.
- The authors reported that some newborns were delivered by caesarean section. Compared to spontaneous delivery, this may induce alterations or introduce a new variable. Are there any indications in this regard? Could anesthesia have modified and/or influenced serum amino acid levels?
Answer: Thank you for your very pertinent and insightful question. During the research methodology design phase, we were aware that we would encounter this issue. However, to avoid reducing the sample size, we decided not to exclude women in regular labor and instead applied a pair-matching method.
Approximately half of the patients were treated with insulin for GDM. Are there differences between treated and untreated patients in terms of blood parameters and measured variables?
Answer: Type of therapy (insulintherapy or diet therapy) have very low effect on children’s weight (Figure 12). In case of amino acid profiles, glycine from venous serum has significantly higher level in case of insulin therapy (p = 0.04), but in both cases median level of glycine is 0: threonine from amniotic fluid has significantly higher level in case of insulin therapy (FC = 1.5, p = 0.01)
- How does the difference in weight affect the observed differences? Body weight and BMI should also be included in the correlations to assess the dependence of the measured variables on these two parameters.
Answer: Mother’s BMI was the most important parametr for GDM development and medium-importance parametr the newborn’s weight. (Figure 11, Figure 12). Also, BMI demonstrated weak negative correlations with proline (r = -0.33, p = 0.004), alanine (r = -0.26, p = 0.02), histidine (r = -0.28, p = 0.02), and γ-aminobutyric acid (r = -0.25, p = 0.03) (Figure 13)
- Other minor revisions: in the tables, some decimals are indicated with a comma instead of a period. Please ensure consistency.
Answer: Tables were corrected
- In the discussion, it would be appropriate to add references regarding the studies cited in lines 400–402 and 425–430.
Answer: References were added